# Influence of Core-Stability Exercises Guided by a Telerehabilitation App on Trunk Performance, Balance and Gait Performance in Chronic Stroke Survivors: A Preliminary Randomized Controlled Trial

**DOI:** 10.3390/ijerph19095689

**Published:** 2022-05-07

**Authors:** Carina Salgueiro, Gerard Urrútia, Rosa Cabanas-Valdés

**Affiliations:** 1Physiotherapy Department, Faculty of Medicine and Health Science Campus Sant Cugat, Universitat Internacional de Catalunya, 08195 Barcelona, Spain; rosacabanas@uic.es; 2Institut d’Investigació Biomèdica Sant Pau (IIB Sant Pau), CIBER de Epidemiología y Salud Pública (CIBERESP), 08025 Barcelona, Spain; gurrutia@santpau.cat

**Keywords:** core-stability exercise, trunk, balance, gait, stroke, telerehabilitation

## Abstract

Stroke is one of the main causes of disability. Telerehabilitation could face the growing demand and a good strategy for post-stroke rehabilitation. The aim of this study is to examine the possible effects of therapeutic exercises performed by an App on trunk control, balance, and gait in stroke survivors. A preliminary 12-week randomized controlled trial was developed. Thirty chronic stroke survivors were randomly allocated into two groups. Both groups performed conventional physiotherapy, in addition to, the experimental group (EG) had access to a telerehabilitation App to guide home-based core-stability exercises (CSE). Trunk performance was measured with the Spanish-Trunk Impairment Scale (S-TIS 2.0) and Spanish-Function in Sitting Test. Balance and gait were measured with Spanish-Postural Assessment Scale for Stroke patient, Berg Balance Scale and an accelerometer system. In EG was observed an improvement of 2.76 points in S-TIS 2.0 (*p* = 0.001). Small differences were observed in balance and gait. Adherence to the use of the App was low. CSE guided by a telerehabilitation App, combined with conventional physiotherapy, seem to improve trunk function and sitting balance in chronic post-stroke. Active participation in the rehabilitation process should be increased among stroke survivors. Further confirmatory studies are necessary with a large sample size.

## 1. Introduction

Stroke is the most frequent cerebrovascular disease and is the leading cause of adult disability in the developed countries [1,2]. Sensory and motor deficits derived from neurological injury, such as paralysis, impaired balance or spasticity, are the most common impairments and are the basis of the resulting degree of physical disability [3]. The affectations not only come out in face and in upper and lower limb, but also in the trunk. Trunk performance plays an important role in biomechanics of human movement and is closely related to lower limb function, balance and gait [4,5]. For this reason, in physiotherapy sessions, improve the trunk function of stroke patient is essential, in all phases of rehabilitation [6,7].

The practice of specific lumbopelvic stability exercises, known as core-stability exercises (CSE), incorporate the direct activation of muscle groups of the trunk and improving their performance. This approach is recommended in the guidelines of good clinical practice, and it is more effective than conventional therapy not only in the recovery of trunk control but also exerts a great influence on the balance and gait [8,9,10,11,12,13,14]. In the recent review by Gamble et al. [15], the authors conclude that the addition of CSE to usual care after stroke can improve trunk control and dynamic balance.

Strategies must be found to introduce the practice of personalized exercises, as CSE, in the daily life of individuals. Telerehabilitation can be a good plan of action for this purpose. This rehabilitation modality could reduce costs in patient assistance, as transfers, human and material resources. Telerehabilitation can also contribute to equalize healthcare access, especially in rural areas. It could also be used to prevent deterioration of the health condition or to detect unfavorable situations outside the acute healthcare phase [16,17]. Creating exercise groups can be used to reduce costs in human resources, however patient care is no longer personalized. The introduction of telerehabilitation in the process of recovery and follow-up of stroke survivors may be a solution to the growing demand and has positive results related to the satisfaction of the participants [18]. Providing immediate home rehabilitation services through telerehabilitation would ensure continuous monitoring of patients and improve not only their state of health but even their Quality of Life [18]. This need has been prioritized during lockdown due to the COVID-19 epidemic [19,20].

It is possible to find neurorehabilitation Apps such as language rehabilitation, cognitive disorders and incentives to physical activity [21,22,23]. However, most of the apps for self-management or sensorimotor stroke recovery are incomplete, depersonalized, and their results are not clear [23,24,25,26]. Home-based trunk training is effective in improving trunk muscle strength, functional sitting range and trunk motor control after stroke in subjects without somatosensory deficits [27]. However, there are currently no references on the practice of CSE at home guided and monitored by an App. Therefore, a preliminary study was carried out to evaluate the effectiveness of CSE guided by a telerehabilitation application on Quality of Life, participation in activities of daily living and functionality of chronic stroke survivors. Trunk, balance and gait performance were also assessed. This study aims to find out if CSE performed via an App can have possible positive effects on trunk control, sitting and standing balance and gait performance in chronic stroke survivors as a study of the integrity of the protocol.

## 2. Materials and Methods

### 2.1. Study Design, Ethics and Sample Characteristics

This preliminary trial was a prospective single-blinded, prospective, randomized, controlled trial. The study protocol was approved by the Clinical Research Ethics Committee of the Universitat Internacional de Catalunya (FIS-2020-01) and registered in the ClinicalTrials.gov database (NCT04477252). The study followed the research principles of Declaration of Helsinki 2013.

As this is a novel and preliminary study, and is limited by the COVID-19 pandemic, no sample study calculation has been performed and the minimum number of 30 participants was established. This study was carried out during the COVID-19 pandemic in which stroke patient care was limited. All eligible participants were recruited from the Neurorehabilitation Clinic, a procedure that limited the loss of trial subjects. The inclusion criteria for this study included medical diagnosis of stroke with cortical or subcortical, ischemic or hemorrhagic involvement with more than 6 months of recovery, clinical symptoms of hemiplegia or hemiparesis, being over 18 years of age, have the ability to understand and execute simple instructions, a score equal to or less than 10 in the Spanish version of the Trunk Impairment Scale 2.0 (S-TIS2.0) [28,29] and be a frequent user of smartphones or tablets (a family member or caregiver could be considered). The exclusion criteria were presence of any neurological or neuromuscular disease or worsening of any of the comorbidities, to suffer another episode of stroke and fractures or important structural alterations in any of the lower limbs (e.g., orthopedic problem of the lower limbs). Individuals with aphasia but capable of understanding and executing simple commands were included with prior consultation with the respective neuropsychologist or speech therapist. Likewise, the caregiver was considered as the user of the App.

Randomization, using random.org program, was carried out by the main researcher of the study, assigning a value of 1 to the experimental group (EG) and a value of 2 to the control group (CG) in a 1:1 ratio.

Participants who fulfilled the inclusion criteria were informed about the study by the main researcher and signed the informed consent. Both groups received conventional physiotherapy, and, in addition, the EG (n = 15) performed CSE by Farmalarm App. The evaluator physiotherapist was unaware of the intervention group of the participants and all the data collected were blinded to the principal investigator, who controlled the interventions. The interventions have had a total duration of 12 weeks for all participants and 3 data collection periods.

### 2.2. Interventions

Both groups (CG and EG) underwent the conventional physiotherapy according to the recommend by Teasell et al., 2020 [6]. It consisted of one-hour face-to-face session of therapeutic techniques such as muscle stretching to reduce hipertonicity or spasticity, passive and functional mobilization of body segments affected by stroke, practice of sitting and standing posture and gait, task and aerobic training as cycling or treadmill training [6]. The techniques used were chosen at the discretion of the physiotherapist in charge following the clinical practice guidelines. The intervention was totally adapted and personalized to the needs and capacities of the patient. Participants maintained their usual dose of treatment during participation in this study. In accordance with clinical recommendations the mean frequency of the sessions was 1 h two times a week for 12 weeks [30]. The physiotherapy sessions were face-to-face and individualized under the responsibility of a physiotherapist with special training and more than 2 years of work experience in neurorehabilitation. The conventional therapy was under responsibility of Clínica de Neurorehabilitación, respecting the health security protocol adopted to face the COVID-19 pandemic.

The participants of the EG, in addition to conventional physiotherapy, had individual access to the Farmalarm App as a telerehabilitation tool to guide adapted home-based CSE [31]. The Farmalarm App (Inmovens Solution, Barcelona, Spain) was specifically adapted for this study. Previously, it was used to monitor adherence to pharmacological treatment in stroke patients [32]. It can be downloaded for free on Android and IOS, but access is limited by a unique code and password assigned by the main researcher. The CSE program, which is part of the rehabilitation section, was developed and introduced by the researchers. Although the exercises should be personalized for each user, in this study phase the CSE guide introduced in the App was common [11,12]. Users have been able to voluntarily access the exercises guide (description, photo and video) and to confirm its performance. Participants were asked to perform 10 repetitions of each of the 32 exercises proposed in the program and were encouraged to perform as many exercises as possible, respecting their perception of tiredness, taking as many breaks as they found necessary. The exercises were introduced in order of difficulty, from the supine position to a seated position on an unstable base. Although the exercises were always presented in the same order, the participants were free to navigate through the menu of the exercises choosing the order they preferred or skipping the exercises that they could not perform or did not feel safe to do so. Figure 1 shows the proposed CSE program to be carried out at home with the help of the App for 12 weeks, 5 days a week.

The main researcher has carried out a face-to-face session of initiation training to use of the App (explanation and a short practice of the proposed exercises) and had access to the administration panel of the App for individual monitoring of each user. The participants were contacted by phone on a regular basis to ensure that they did not have problems with the use of the App.

### 2.3. Outcome Measures

The measure outcomes were trunk performance and sitting balance, assessed using S-TIS2.0 [28,29] and the Spanish version of Function in Sitting Test (S-FIST) [33,34,35]. Standing balance was assessed by Spanish version of Postural Assessment Scale for Stroke Patients (S-PASS) [36,37], by Berg Balance Scale (BBS) [38] and the number of falls registered. G-Walk accelerometer system from BTS Bioengineering [39] was used to assess gait parameters. This gait analysis system allows the collection of several gait parameters objectively (e.g., step length, speed, cadence and symmetry) in a single test and, seems to be a good assessment tool with stroke patients [40,41]. Adherence to telerehabilitation as an exercise guide and the number of exercises performed daily has been recorded in the control panel of the App. The assessments were carried out in a single session by a physiotherapist expert in neurorehabilitation, who was unaware of the group to which the patient had been assigned. The interval between assessment times, baseline (T0), 6 (T1) and 12 (T2) weeks were respected. In order to respect the protocol against the COVID-19 epidemic, phone or video-calls were made in case of not being able to carry out face-to-face assessments. The outcomes that could not be assessed, were considered absent and have not been taken into account in the statistical analysis.

### 2.4. Statistical Analysis

Descriptive statistics have been used for the characterization of the sample. The mean value and standard deviation were calculated for continuous data in both groups and individual’s characteristics were described using frequencies and percentages. Statistical analysis was performed with IBM SPSS Statistics software (Version 24) and R Project software [42]. The parametric analysis was performed, and the Shapiro–Wilk test was used to verify the distribution of the variables. The difference between the pre- and post-intervention value (ΔT0–T2) was studied in order to facilitate the interpretation of the results. Paired-samples t-test was used to study changes from baseline to the end of the intervention and independent-samples *t*-test was used to analyses changes between groups. Two-sided of *p*-value  < 0.05 were considered statistically significant. Absolute and relative values have been taken into account to study the usability of the telerehabilitation App Farmalarm.

As this was a preliminary study an analysis of the measure of change over time was not considered relevant.

## 3. Results

Thirty-two participants were recruited from Clínica de Neurorehabilitación in Barcelona, from May 2020 to May 2021. The 30 participants who met the inclusion criteria, were allocated in the CG (n = 15) and EG (n = 15) (Figure 2). Although small differences were observed between the groups in terms of age, number of risk factors and side of the body affected, the differences were not significant. (Table 1). In the CG, the mean age was higher than in the EG but showed less variation. The CG participants presented, on average, a greater number of risk factors, representing a sample with a more precarious health condition than that of the EG. Relatively to the affected body side, in the CG there was a predominance of participants with left hemiplegia or hemiparesis while in the EG there was a predominance of right hemiplegia or hemiparesis.

Table 2 presents the results of trunk function measures, where the *p*-value reflects the significance of the differences obtained intragroup and between groups pre- and post-intervention. According to the analysis of sitting balance by S-TIS 2.0, EG was observed to be superior to CG in balance and total score, whereas for coordination a significant difference at the end of intervention was observed only in the EG although. Specifically, between the beginning and the middle of the study, was observed an increase of 1.71 points, and between the beginning and end of this study, was observed an increase of 2.76 points. In contrast, no significant improvement was observed in the CG in any of these outcomes. In relation to the balance in sitting position, measured with the S-FIST, no changes were observed.

Table 3 presents the results of standing balance, where the *p*-value reflects the significance of the differences obtained intragroup and between groups pre- and post-intervention. In the first section of the S-PASS test, corresponding to the assessment of mobility, improvements were observed in both groups but only, the rise of 1.5 points observed in EG had statistical significance in intragroup comparison. No significant differences were observed between interventions. In the balance section of this assessment test, no significant changes were observed in each group separately or in the comparison between groups. In the total score of the S-PASS test, no statistical significance was observed between groups.

In BBS, also used to assess balance, an increase in the score has been observed in both groups. This difference had statistical significance in intragroup comparison but did not achieve statistical significance in comparison between groups.

Relative to the number of falls, no differences have been observed. The rise of number of falls in CG is due to a participant who started to use a bicycle outdoor.

The gait parameters have been studied and analyzed individually (Table 4). Two participants in this study were not capable of walking at the beginning of the study, one from the CG and the other from the EG. The statistical analysis of means and variances was carried out on 14 individuals in each group. No obtained data was taken into account as omission. The difference obtained between the evaluation values and the normal or expected values has been calculated using the G-Walk system software data. Higher results represent a greater deviation compare with the expected data, and a worse result and values close to zero represent a small deviation and a better result. About stance and swing phase, the expected values were 57–61% and 36.5–43.6%, respectively. Major deviations were observed in swing phase of the affected limb in both groups, and the bigger improvement was observed in this parameter in EG.

About gait support, the expected value for double support of the stance phase was 7.2–13.4% and the expected value for single support was 36.3–41.4%. The most affected parameter was single support of the affected limb of EG. Plus, the bigger improvement was observed in double support of the affected limb in EG (Table 5).

Gait cadence, measured in steps per minute, gait speed, measured in meters per second, and stride, measured in meters, varies from individual to individual because it depends on anthropometric characteristics. For this reason, the study was carried out with the value of the difference between the data obtained and the expected values. No regular results were obtained.

In relation to the length of the step, where is expected the symmetry between the right and left limbs (50%/50%), small variations were observed, in this case, between affected and less affected limbs. At baseline, an asymmetry of 52.72–47.28% was observed in the control group and 55.32–44.68% in the experimental group (affected limb-less affected limb). At the end of the study, asymmetry was maintained in the control group at 52.74–47.26% and was reduced to 51–67–48.33% in the experimental group.

Adherence to the use of the App to perform CSE at home, was monitored by the App’s administration panel. This panel recorded the time of use of the App and the exercises consulted and marked as carried out. Adherence has been low, with an average use of 13.66%. On average, the patients performed 12 exercises for each connected day, which corresponds to 37.09% of the total of the exercise program proposed to be performed 5 days/week (Table 5).

In summary, in the analysis of the different measure outcomes of this study, significant changes were observed in the EG in trunk function measured by S-TIS2.0. Improvements were observed in the other outcomes of this preliminary study in the EG, but without statistical significance.

## 4. Discussion

CSE exercises by means of Farmalarm App in addition to conventional physiotherapy seems to improve trunk function and sitting balance in chronic post-stroke. This preliminary study illustrated the utility of the telerehabilitation App as a guide to CSE, whose effectiveness is already known [43]. In a recent review by Cabrera-Martos et al. [44], the authors conclude that CSE, performed alone or in combination with other physiotherapy techniques, are effective in relation to trunk performance. Similar results were found in this preliminary study in which Farmalarm App was used to administer the CSE. The greatest change was observed in trunk performance or sitting balance, directly related to trunk function. These results also reinforce the introduction of the trunk approach as CSE in the recovery process of stroke survivors [6,7].

Physiotherapy is essential for people with disabilities and the use of telerehabilitation as a physiotherapy method seems to have several benefits [16,17]. It can be useful in situations such as confinement during the COVID-19 health crisis [45]. The use of the telerehabilitation App to perform CSE seems to bring benefits to users with regard to standing balance, but the results are not obvious. About gait, the results were very dispersed in both groups, and it is recommended to add other assessment methods to the gait analysis for future studies. No clear results have been obtained in this measure outcomes, but previous studies establish the relationship between the trunk and balance and gait [4,5]. In the last review about the efficacy of CSE in stroke survivors [15], there is evidence about these therapeutic exercises on gait speed but not on other gait parameters.

About trunk function, standing balance and gait, similar results were found in a study with individuals with stroke in the sub-acute stage [46]. It leads to the conclusion that CSE seem to be more effective in relation to trunk function and balance in sitting position at all stages of post-stroke rehabilitation.

The lack of greater results on this preliminary study may be related to poor adherence to performing exercises autonomously at home. This was the first study in which adherence levels have been objectively measured, and according to an interview at the end of the study, the lack of adherence may be related to a high demand for using the App in this study and the high number of exercises proposed to be performed in each session. However, some participants have completed all 32 proposed exercises in a single session. In addition, technical challenges and access problems may pose potential barriers to telerehabilitation as a service delivery model [47]. However, resistance to telerehabilitation is expected to decrease due to the introduction of new technology in daily life in modern society. The participants who had access to telerehabilitation were contacted periodically by phone call to guarantee the absence of problems with the use of the App. This continuous contact may have influenced the participants’ regular physical activity. On the other hand, the participants could have memorized the exercises and performed them without using the App. Future studies with a shorter therapeutic exercise program and a lower weekly frequency are recommended. It is thought that these small intervention changes will make it easier to introduce telerehabilitation into the daily routine of stroke patients.

Telerehabilitation seems to be as effective as face-to-face therapy [48,49]. However, the levels of adherence are different. Saywell et al. [50], conclude that telerehabilitation can be effective in maintaining the physical condition of users and preventing deterioration. This aspect is important for stroke survivors due to the chronicity of the dysfunction and secondary complications. Furthermore, telerehabilitation should not completely replace face-to-face therapy due to the good results and satisfaction of stroke survivors relative to the direct interaction with therapists [51].

In the last year there was a significant increase in publications related to telerehabilitation in general and in stroke survivors [52]. It seems that the health crisis experienced due to the COVID-19 pandemic aroused the interest in this area. In 2018 there were 33 indexed research papers in PubMed library, and in 2021 there were 83 [52]. In the latest systematic review on telerehabilitation in stroke patients, the authors state that studies regarding acceptability and reliability are important for the study of telerehabilitation [53]. This study provides information to measure the effect of telerehabilitation, and relevant information for the future introduction of telerehabilitation in clinical practice with patients with post-stroke disorders.

Stroke survivors’ self-responsibility behaviors should be reinforced in their recovery process. One of the strategies could be to clarify to the patients and their families the relationship between the recommended exercises and the improvement of functional prognosis.

## 5. Limitations

Firstly, this preliminary study did not have a sample size calculation, which limits the scope of interpretation of the results. However, it is exploratory in nature regarding the possible therapeutic impact and the acceptability. This study provides relevant base information for future clinical trials. Low adherence made it difficult to validate the results. However, the results suggest clinical changes.

For this publication, the authors have focused on the secondary measure outcomes of the study registered in Clinicaltrials.gov.

## 6. Conclusions

CSE guided by a telerehabilitation app, combined with conventional physiotherapy, are feasible and seems to improve trunk function and sitting balance in patients with chronic stage stroke. Although CSE seem to improve standing balance, no significant results were obtained. To measure the effects on gait, other methods of analysis must be incorporated. Adherence to telerehabilitation, as a guide tool for performing exercises autonomously at home, seems to be low. Future studies are needed to find out the reasons for low adherence to the use of telerehabilitation. Strategies must be found to increase self-responsibility and involvement in the recovery process by survivors and their families. Strategies to improve access and adherence to telerehabilitation as a guide to specific and personalized therapeutic exercises for stroke patients should also be studied.

## Figures and Tables

**Figure 1 ijerph-19-05689-f001:**
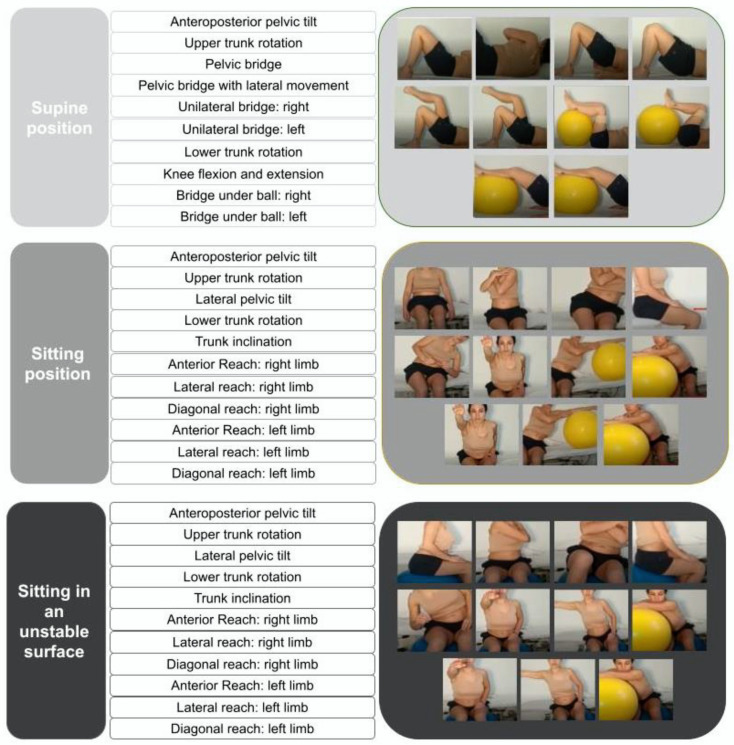
Core-stability exercise protocol for stroke patients included in the telerehabilitation App.

**Figure 2 ijerph-19-05689-f002:**
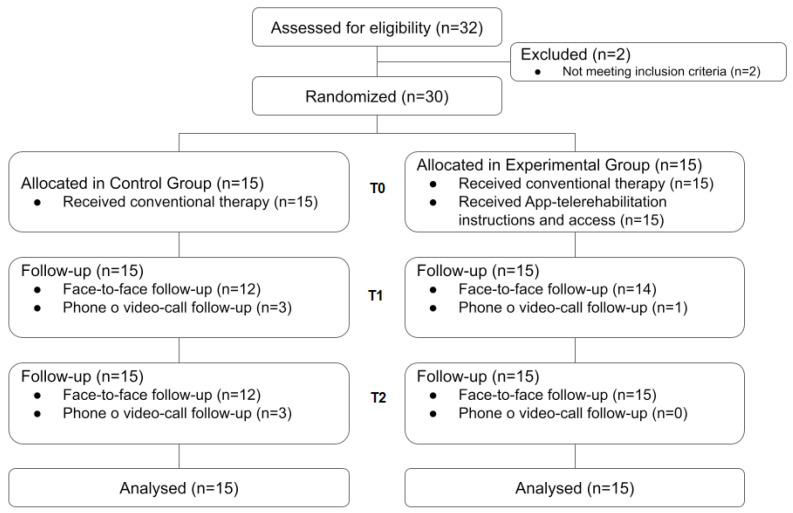
Flowchart of the study.

**Table 1 ijerph-19-05689-t001:** Characteristics of the experimental and control group at baseline.

Variables	Pre-Intervention	
	Control Group (n = 15)	Experimental Group (n = 15)
**Characteristics**		
Age (years), mean (SD)	64.53 (9.40)	57.27 (14.35)
Weight (Kilograms), mean (SD)	75.83 (12.76)	80.83 (11.05)
Height (centimeters), mean (SD)	164.40 (10.18)	169.13 (9.49)
Body Mass Index, mean (SD)	27.73 (3.54)	26.15 (3.61)
Sex (male/female)	10/5	10/5
Stroke risk factors (yes/no)	8/7	4/11
Predominant risk factor	High Blood Pressure	High Blood Pressure
Active lifestyle before stroke (yes/no)	7/8	8/7
Stroke (hemorrhagic/ischaemic)	5/10	5/10
Predominant stroke localization	Middle Cerebral Artery	Middle Cerebral Artery
Hemiplegia (right/left)	7/8	10/5
Time post-stroke (years), mean (SD)	4.06 (4.43)	4.61 (3.38)
Smartphone user (yes/no)	14/1	15/0
**Coventional Physiotherapy**		
Times/week (frequency)		
>4	1	0
3–4	0	2
1–2	13	12
no therapy	1	1
Scales		
**Sitting balance, mean (SD)**		
S-TIS 2.0 (balance)	4.27 (1.62)	4.73 (2.12)
S-TIS 2.0 (coordination)	3.07 (1.33)	2.87 (1.13)
S-TIS 2.0 (total)	7.33 (2.38)	7.60 (2.77)
S-FIST (total)	56.86 (3.40)	52.6 (9.58)
**Standing balance, mean (SD)**		
S-PASS (mobility)	18.67 (2.85)	18.93 (2.89)
S-PASS (balance)	10.47 (2.61)	9.46 (2.23)
S-PASS (total)	29.13 (5.00)	28.4 (4.90)
BBS	41.27 (15.42)	43.2 (12.73)
Falls	0	0
**Gait—G-Walk, mean (SD)**		
Stance phase % (affected limb)	5.18 (4.36)	6.13 (4.70)
Stance phase % (less affected limb)	5.38 (5)	5.13 (3.72)
Swing phase % (affected limb)	8.11 (8.03)	10.05 (9.10)
Swing phase % (less affected limb)	7.13 (6.71)	7.54 (8.20)
Double Support % (affected limb)	4.56 (5.28)	4.51 (5.84)
Double Support % (less affected limb)	4.50 (4.89)	1.57 (1.33)
Single Support % (affected limb)	5.68 (5.18)	6.20 (4.39)
Single Support % (less affected limb)	5.86 (4.75)	4.85 (4.18)
Cadence (steps/minute)	30.47 (20.01)	22.49 (23.07)
Speed (meters/seconds)	0.39 (0.38)	0.36 (0.35)
Stride (meters), (affected limb)	0.33 (0.28)	0.35 (0.40)
Stride (meters), (less affected limb)	0.34 (0.28)	0.36 (0.40)

Values are presented as mean (SD) or absolute frequency. BBS (Berg Balance Scale); S-FIST (Spanish version of Function in Sitting Test); S-PASS (Spanish version of Postural Assessment Scale for Stroke Patients); S-TIS 2.0 (Spanish version of the Trunk Impairment Scale 2.0); SD (standard deviation).

**Table 2 ijerph-19-05689-t002:** Trunk function measures, sitting balance middle and post-intervention.

Sitting Balance	Control Group			Experimental Group			
	Middle Intervention	Post-Intervention	ΔT0–T2	Middle Intervention	Post-Intervention	ΔT0–T2	*p*-Value (Intergroup *)
S-TIS 2.0 (balance)	4.15 (2.54)	4.31 (1.84)	0.23 (1.30)	6.15 (2.12)	6.71 (2.33)	1.86 (1.56)	0.007
*p*-value (intragroup *)			0.534			**0.001**	
S-TIS 2.0 (coordination)	3.00 (1.91)	3.15 (1.91)	0.08 (1.44)	3.15 (1.72)	3.64 (1.15)	0.71 (0.91)	0.424
*p*-value (intragroup *)			0.851			**0.012**	
S-TIS 2.0 (total)	7.15 (4.20)	7.46 (3.57)	0.31 (2.10)	9.31 (3.5)	10.36 (3.03)	2.57 (1.87)	0.032
*p*-value (intragroup *)			0.606			**0.000**	
S-FIST (total)	54.15 (3.87)	53.54 (6.84)	−1.15 (4.36)	52.69 (10.26)	54.71 (3.47)	2.36 (6.66)	0.574
*p*-value (intragroup *)			0.358			0.208	

Values are presented as mean (SD). * pre and post-intervention values. *p*-value (statistical significance); SD (standard deviation); S-FIST (Spanish version of Function in Sitting Test); S-TIS 2.0 (Spanish version of the Trunk Impairment Scale 2.0); ΔT0–T2 (intragroup improvement).

**Table 3 ijerph-19-05689-t003:** Balance outcome measures, standing balance middle and post-intervention.

Standing Balance	Control Group			Experimental Group			
	Middle Intervention	Post-Intervention	ΔT0–T2	Middle Intervention	Post-intervention	ΔT0–T2	*p*-Value (Intergroup *)
S-PASS (mobility)	18.85 (3.31)	18.69 (3.50)	0.15 (1.52)	19.54 (3.13)	20.43 (3.48)	1.43 (2.28)	0.208
*p*-value (intragroup *)			0.721			0.035	
S-PASS (balance)	10.46 (2.70)	10.46 (3.01)	−0.08 (1.04)	9.69 (2.39)	9.79 (2.52)	0.29 (0.91)	0.532
*p*-value (intragroup *)			0.794			**0.263**	
S-PASS (total)	29.31 (5.54)	29.15 (6.04)	0.08 (1.93)	28.54 (5.43)	30.21 (5.35)	1.71 (2.43)	0.633
*p*-value (intragroup *)			0.888			**0.02**	
BBS	40.69 (15.69)	42.54 (14.49)	2.46 (2.85)	44 (13.22)	44.93 (12.29)	1.93 (2.95)	0.647
*p*-value (intragroup *)			**0.009**			**0.029**	
Falls	0.47 (0.88)	0.15 (0.38)	0.15	0.15 (0.38)	0.14 (0.36)	0.14	0.999
*p*-value (intragroup *)			0.165			0.165	

Values are presented as mean (SD). * pre and post-intervention values. BBS (Berg Balance Scale); *p*-value (statistical significance); SD (standard deviation); S-PASS (Spanish version of Postural Assessment Scale for Stroke Patients); ΔT0–T2 (intragroup improvement).

**Table 4 ijerph-19-05689-t004:** Gait analysis middle and post-intervention.

Gait	Control Group			Experimental Group			
	Middle Intervention	Post-Intervention	ΔT0–T2	Middle Intervention	Post-Intervention	ΔT0–T2	*p*-Value (Intergroup *)
**Phase**							
Stance phase % ** (affected limb)	4.53 (2.99)	4.01 (2.21)	1.74 (5.14)	5.25 (3.25)	6.43 (4.21)	0.16 (6.13)	0.568
*p*-value (intragroup *)			0.541			0.833	
Stance phase % ** (less affected limb)	4.07 (3.03)	4.95 (2.99)	1.14 (5.78)	5.19 (4.97)	6.53 (4.63)	−0.93 (5.23)	0.395
*p*-value (intragroup *)			0.334			0.334	
Swing phase % ** (affected limb)	5.00 (5.22)	6.13 (4.60)	1.98 (8.36)	5.87 (6.54)	8.33 (8.19)	1.71 (9.85)	0.909
*p*-value (intragroup *)			0.800			0.648	
Swing phase % ** (less affected limb)	3.71 (3.99)	4.99 (4.04)	2.14 (7.81)	5.35 (6.45)	7.02 (6.30)	0.52 (7.76)	0.923
*p*-value (intragroup *)			0.881			0.848	
**Support**							
Double Support % ** (affected limb)	2.65 (3.92)	2.77 (3.47)	1.79 (4.65)	0.82 (0.97)	1.27 (1.76)	0.30 (2.51)	0.299
*p*-value (intragroup *)			0.512			0.096	
Double Support % ** (less affected limb)	3.82 (9.52)	2.68 (3.36)	1.82 (6.38)	2.49 (3.17)	4.56 (3.95)	−0.05 (7.06)	**0.034**
*p*-value (intragroup *)			0.370			**0.022**	
Single Support % ** (affected limb)	5.16 (6.86)	5.3 (2.79)	0.56 (5.98)	4.23 (3.46)	6.03 (4.98)	−1.18 (6.53)	0.921
*p*-value (intragroup)			0.498			0.405	
Single Support %** (less affected limb)	5.16 (6.86)	5.30 (2.79)	0.56 (5.98)	4.23 (3.46)	6.03 (4.98)	−1,18 (6.53)	0.840
*p*-value (intragroup *)			0.117			0.428	
**Others**							
Cadence (steps/minute **)	30.88 (21.93)	26.88 (24.97)	7.44 (16.74)	32.9 (22.19)	29.49 (21.68)	−4.89 (18.57)	0.292
*p*-value (intragroup *)			0.606			0.929	
Speed (meters/seconds)	0.86 (0.44)	0.94 (0.35)	0.01 (0.05)	0.95 (0.34)	0.94 (0.36)	−0.08 (0.45)	0.951
*p*-value (intragroup *)			0.940			0.929	
Stride (meters), (affected limb)	1.20 (0.50)	1.26 (0.31)	0.01 (0.15)	1.28 (0.70)	1.30 (0.55)	−0.06 (0.52)	0.984
*p*-value (intragroup *)			0.55			0.606	
Stride (meters), (less affected limb)	1.21 (0.50)	1.26 (0.32)	−0.02 (0.14)	1.30 (0.71)	1.30 (0.56)	−0.06 (0.52)	0.987
*p*-value (intragroup *)			0.547			0.615	

Values are presented as mean (SD). * pre and post-intervention values. ** difference between normal values and measurements. *p*-value (statistical significance); SD (standard deviation); ΔT0–T2 (intragroup improvement).

**Table 5 ijerph-19-05689-t005:** Telerehabilitation app adherence.

	Adherence		Exercises Performed	
		(%)	Exercises/Day	(%)
Mean (SD)	8.2 (8.1)	13.66	11.84 (9.41)	37.09
Min-max	0–23		0–32	

## Data Availability

Data is contained within the article.

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
