# Peer review of "Influence of Core-Stability Exercises Guided by a Telerehabilitation App on Trunk Performance, Balance and Gait Performance in Chronic Stroke Survivors: A Preliminary Randomized Controlled Trial"

_ijerph, 2022, doi:10.3390/ijerph19095689_

Round 1
Reviewer 1 Report
Dear authors, the manuscript has improved but there are still concerns to be filed .. Describing the intervention of the control is decisive, greater methodological rigor is essential
144 Enrich the figure legend with the path followed .. 10 repetitions, up to what the patient is able to do .. which is the intervention of the experimental group ..
I cannot understand what is meant by usual care of the control group, because if by usual care we do not mean a rehabilitation protocol, we are evaluating the effect of rehabilitation in the telemedical context versus no rehabilitation intervention. In these subjects it is clear a priori that whatever the scenario set up, rehabilitation is effective compared to usual care..
L 146 For this publication, and respecting the aim of the study, balance and gait were 146 considered as the primary outcome
But in the abstract you said: Secondary outcomes were balance and gait
170 change score? Mean difference?
is the control significantly older?
Is it only in the results section that we start talking about middle of the study? what does it mean? describe the timepoints in the methods and repeat them in the results ... consider the intra-group improvement as ΔT0-T1
The type of score does not fit much, due to the ability to reduce or increase by 2 or 3 points, even if only based on the symmetry of the rotation of the trunk. The percentage terms of modification are superfluous, instead of the change score I would suggest the MD and the standard error.
The PASS is much more straightforward and comparable, as for the tabular representation suggestions above
Figure 3 unscientific, insert the SD
I suggest removal of figure 4
Author Response
144 Enrich the figure legend with the path followed .. 10 repetitions, up to what the patient is able to do .. which is the intervention of the experimental group ..
(L138-140). The requested information has been added
I cannot understand what is meant by usual care of the control group, because if by usual care we do not mean a rehabilitation protocol, we are evaluating the effect of rehabilitation in the telemedical context versus no rehabilitation intervention. In these subjects it is clear a priori that whatever the scenario set up, rehabilitation is effective compared to usual care..
(L107-121). The usual care concept was eliminated so as not to confuse it with medical usual care. Conventional physical therapy is described in the text.
L 146 For this publication, and respecting the aim of the study, balance and gait were 146 considered as the primary outcome
But in the abstract you said: Secondary outcomes were balance and gait
(L143-144) The information was corrected in the text.
170 change score? Mean difference?
'Change scores' are known to provide a simple summary measure of the average change in a variable between two time points and are commonly used when analyzing 'change' in an outcome relative to a reference exposure. Because there are not always significant changes in the variables, we have chosen to use the term "change score". However, the suggestion to use the term ΔT0-T2 was accepted.
is the control significantly older?
(L187-189). No significant differences were found between the groups in any of the variables. Even so, the researchers highlighted the variables that presented small differences.
Is it only in the results section that we start talking about middle of the study? what does it mean? describe the timepoints in the methods and repeat them in the results ... consider the intra-group improvement as ΔT0-T1
(L164). Variable assessment times are described in the methodology and in Figure 1 of the manuscript. The expression ΔT0-T2 was considered as the improvement between the initial and final moment of the study.
The type of score does not fit much, due to the ability to reduce or increase by 2 or 3 points, even if only based on the symmetry of the rotation of the trunk. The percentage terms of modification are superfluous, instead of the change score I would suggest the MD and the standard error.
The percentage of improvement was eliminated and it was confirmed that all the results are presented as mean (standard deviation).
The PASS is much more straightforward and comparable, as for the tabular representation suggestions above
(Table 3). The two sections of the PASS scale and the total are presented. Mean and standard deviation values are presented.
Figure 3 unscientific, insert the SD
Figure 3 summarizes the gait analysis. Standard deviation values have been included.
I suggest removal of figure 4
Figure 4 was removed.
Reviewer 2 Report
In background of the abstract the aim of the study data should appear.
METHODS
The study seems to be a pilot study. In this case, the objective is not to compare treatments groups, and prove the effects and the superiority of the treatment but to test feasibility, trial procedures and processes. A pilot study can be used to evaluate the feasibility of recruitment, randomization, retention, assessment procedures, and implementation of the novel intervention.
As pilot trials do not have the same objectives as the main trial, the sample size calculation is not based on formal power considerations but on the pragmatics of recruitment and the necessities for examining feasibility.
The main and secondary outcomes should be redefined also because in the abstracts line 17 the authors declaire: “secondary outcomes were balance and gait.”. On the contrary in the outcome measures section lines 146-147 they stated “balance and gait were considered as the primary outcome”
Exclusion criteria lines 92-93: Explain how the understanding and execution of commands were verified in aphasic patients
In the RESULTS section, the authors should first describe feasibility and adherence data. Reorder the results accordingly to the methods.
Specify the number of face-to-face assessments or video calls and the number of outcomes not assessed.
Please, increase the discussion.
Author Response
In background of the abstract the aim of the study data should appear.
(L12-13). The purpose of the study was included in the abstract
METHODS
The study seems to be a pilot study. In this case, the objective is not to compare treatments groups, and prove the effects and the superiority of the treatment but to test feasibility, trial procedures and processes. A pilot study can be used to evaluate the feasibility of recruitment, randomization, retention, assessment procedures, and implementation of the novel intervention.
As pilot trials do not have the same objectives as the main trial, the sample size calculation is not based on formal power considerations but on the pragmatics of recruitment and the necessities for examining feasibility.
This study is a preliminary study, as indicated in the title and in the text, which allowed us to obtain valuable information for future studies.
The main and secondary outcomes should be redefined also because in the abstracts line 17 the authors declaire: “secondary outcomes were balance and gait.”. On the contrary in the outcome measures section lines
146-147 they stated “balance and gait were considered as the primary outcome”
The information was corrected
Exclusion criteria lines 92-93: Explain how the understanding and execution of commands were verified in aphasic patients
(L93-94). The information was added.
In the RESULTS section, the authors should first describe feasibility and adherence data. Reorder the results accordingly to the methods.
The results are in accordance with the objectives of the study. Adherence data have been entered as extra information for a better interpretation of the results. This study will lead to several publications, the current manuscript is referring to the results related to trunk control, balance and gait.
Specify the number of face-to-face assessments or video calls and the number of outcomes not assessed.
This information can be consulted in figure 1 of the manuscript.
Please, increase the discussion.
Some extra information was added in the discussion section.
Round 2
Reviewer 1 Report
Thanks to the authors for their efforts in the latest revision, but I need my final concerns addressed to make the manuscript suitable for publication. 109-113 Thank you for giving “conventional” therapy meaning to the control group. However, I ask you for more detail in the description of this intervention. I'm being tedious, but then again it's a gamechanger to figure out the real gain that the experimental group has over the control. passive and functional mobilization of body segments affected by stroke, practice of sitting and standing posture and gait, task and aerobic training as cycling or treadmill training ... It is a massive intervention also compared to the experimental one, well I ask you to tabulate each step, with during and possible break. Also for the reproducibility of the manuscript. Figure SD is by convention a line above each bar.. not a linking points. It’s inappropriate. Maybe, Use a freeware software “Past4” for plots… not MS excel, please. (https://www.nhm.uio.no/english/research/infrastructure/past/ ) 333 missing reference.. ref: http://dx.doi.org/10.1108/JET-11-2020-0047 the beginning of the discussion by convention must be paraphrased the objective of the manuscript 345 remove study, just limitations..Author Response
Gracias por sus comentarios que nos ayudaron a mejorar la escritura del manuscrito. A continuación le contestamos punto por punto.
Thanks to the authors for their efforts in the latest revision, but I need my final concerns addressed to make the manuscript suitable for publication.
109-113 Thank you for giving “conventional” therapy meaning to the control group. However, I ask you for more detail in the description of this intervention. I'm being tedious, but then again it's a gamechanger to figure out the real gain that the experimental group has over the control. passive and functional mobilization of body segments affected by stroke, practice of sitting and standing posture and gait, task and aerobic training as cycling or treadmill training ... It is a massive intervention also compared to the experimental one, well I ask you to tabulate each step, with during and possible break. Also for the reproducibility of the manuscript.
Conventional therapy is described as much as possible. As referred to in different clinical practice guidelines, intervention in physiotherapy with patients with disabilities, in this case due to stroke, must be totally personalized to the patient's abilities. And taking into account the signs and symptoms and in accordance with the defined therapeutic objectives. On the other hand, protocolizing a conventional intervention would be ethically incorrect by depriving the patient of receiving the best treatment for their needs (e.g.: a patient with stroke (A) has spastic equinus foot as the greatest limitation, so the intervention of the conventional physiotherapy should be focused on it. The patient with stroke (B) has as a major limitation the lack of balance due to great hypotonia in hip stabilizers, so the intervention of conventional physiotherapy should be focused on it.
Finally, we argue that this is a pragmatic study, clinical practice and interventions correspond exactly to reality, which is why the text indicates that the dose and type of therapy were not modified for this study.
All patients of the study performed physiotherapy with therapists trained in neurorehabilitation and with a minimum of 8 years of experience, so the treatment performed was at their discretion. All therapists were up to date in terms of scientific evidence and are aware of the published clinical guidelines.
However, other works published in this journal on stroke rehabilitation were consulted. No examples of the description of conventional therapy have been found.
Eg.: https://doi.org/10.3390/ijerph19063381:
conventional physiotherapy is describe as: based on shoulder, elbow, wrist and finger mobilization, strengthening of UE extensor muscles, stretching exercises for UE flexor muscles, exercises to improve motor control of UE
Eg.: https://doi.org/10.3390/ijerph19010224:
conventional physiotherapy is describe as: upper limb and hand movement and sit-to-stand training for 30 min followed by overground training for 30 min, 3 days/week for 8 weeks, by a physical therapist
To improve the description of conventional physiotherapy, the dose of therapy received was added to the table of the characteristics of the participants.
Figure SD is by convention a line above each bar.. not a linking points. It’s inappropriate. Maybe, Use a freeware software “Past4” for plots… not MS excel, please. (https://www.nhm.uio.no/english/research/infrastructure/past/)
An attempt was made to enter the SD values as indicated but it was concluded that the graph was visually imperceptible due to the amount of data. In previous comments, the graph was considered superfluous because the data is presented in tables. For these two reasons, the team decided to remove the graph from the manuscript.
333 missing reference.
The reference is on line 336. For clarity, it was also added on line 333.
ref: http://dx.doi.org/10.1108/JET-11-2020-0047 the beginning of the discussion by convention must be paraphrased the objective of the manuscript
The recommended article was included in the discussion and duly referenced
345 remove study, just limitations.
The subtitle was corrected
Reviewer 2 Report
The methodological part needs revision. I think I have not explained it well in the previous comment. A preliminary study may be a feasibility study or a pilot study. The objectives should not be the same as the final RCT study in both cases. So, I suggest modifying the goals and reordering the results.
Author Response
Gracias por sus comentarios que nos ayudaron a mejorar la escritura del manuscrito.
The methodological part needs revision. I think I have not explained it well in the previous comment. A preliminary study may be a feasibility study or a pilot study. The objectives should not be the same as the final RCT study in both cases. So, I suggest modifying the goals and reordering the results.
The presented study aims to test the integrity of the study protocol. The information obtained and conclusions obtained are relevant for future studies (l 357). For this reason, and consulting publications on pilot or preliminary study methodology*, the measures outcomes and their order were maintained. However, the main and secondary outcome terms were removed to avoid understanding this study as a confirmatory study (l 151). The aim of the study was also outlined so that it is understood that it is not a confirmatory study (l 69).
*
Gustavo Díaz M. Metodología del estudio piloto. Rev Chil Radiol 2020, 26(4), 172-176.
Moore C, Carter R, Nietert P, Stewart P. Recommendations for Planning Pilot Studies in Clinical and Translational Research. Clin Trans Sci 2011; 4(5), 332–337.
In J, Introduction of a pilot study. Korean Journal of Anesthesiology 2017, 70 (6), 601-605.
This manuscript is a resubmission of an earlier submission. The following is a list of the peer review reports and author responses from that submission.
Round 1
Reviewer 1 Report
The study is about the use in add-on of the telerehabilitation App to improve balance trunk and gait on stroke survivors . Despite to interesting and current topic (the telerehabilitation) and considering it is an RCT study, I report weighty concerns especially related to methodology and analyses plan. The principal points of weakness are:
-the small sample size. Concerning this point, would be more appropriate to consider the results as preliminary or the study as a pilot.
- the aim of the study and the primary outcomes declared in the Clinical Trials are different from those reported in the paper.
Introduction
The authors could go into detail about the aspects of novelty and the relevance of the study concerning the recent literature. The authors could deepen the possible advantages of telerehabilitation for stroke patients and the recent results. The aim of the study is different from the aim declared in the Clinical Trials. The authors could explain this change?
Methods
The authors declared that sample calculation was not performed, they explained that is the first research and they intended to recruit the maximum number of participants. This point is unclear to me. This study is a pilot or a preliminary study of the major study? The sample size is an important feature of any empirical study in which the goal is to make inferences about a population from a sample. Please, the author could explain this point. The outcome measures reported in the paper are incomplete concerning clinical trials. Moreover, the primary outcome was adherence to telerehabilitation APP, instead, the trunk and balance are considered secondary outcomes. The authors could explain this change. The presence of aphasia or the comprehension deficit were considered as exclusion criteria? considering that patients recruited presented left brain damage and it could be interfering with the correct use of the APP
Statistical analysis
The parametric analyses were performed, the normality of the variables was previously verified? The authors could consider performing analysis considering the measure of changing over the time calculating the DELTA between the several timepoints?.
Results
In table 7, 5 and 6 the results could be shown by the graphs. The significant results could be reported in bold.
Discussion
In the discussion, the first result discussed is the feasibility of APP, but this is not the aim of the study or the main result. I suggest discussing the main result that corresponds to the aim of the study reported in the paper. The authors reported "that the study not only focuses the use of telerehabilitation but also on its content", this point is not clear, what the authors would communicate? A limitations section should be inserted.
Author Response
Thank you for your comments, which have helped improve this manuscript. We hope that the answers clarify the doubts raised.
This study is considered a preliminary and novel study for the introduction of telerehabilitation of stroke patients through an App. The small sample is justified by several factors (p. 75).
For this publication, the authors focused on the objectives and secondary variables reported in the study registry (p. 308).
Patients with aphasia were included in this study as long as they preserved the ability to understand and follow simple commands. Likewise, the direct caregiver was considered the user of the App and the person responsible for its use (p.85, 89).
The normal distribution of the variables was confirmed before the statistical analysis (p. 158). However, the calculation of the variation over time with the DELTA calculation has not been considered relevant for this preliminary study (p. 165).
The results with statistical significance were highlighted in the text and a graph of the results of the gait analysis was introduced for better interpretation (p. 234).
The discussion section has been written to be in accordance with the main objective of this publication (p.262) and the study limitations section has been introduced (p.302).
Reviewer 2 Report
The manuscript is interesting, above all because beyond the use of tele-rehabilitation technologies it is one of the first papers that underlines the weight and value of trunk control as a litmus test of autonomy and quality of life.
42 I suggest such a statement with reference to: "Telerehabiliation might provide specific immediate home rehabilitation services, guaranteeing continuous monitoring of patients and improving not only the state of health but above all the quality of life."
(ref: https://doi.org/10.1108/JET-11-2020-0047 )
53 I would suggest enriching the rationale with the lack of focus in the literature on home-based trunk therapy in people with stroke
83 e Figure 1.. The fact that the enrollment is of 30 patients is a result, as well as the described flowchart is a result that led to the start of the study. To this continuation on line 79 add allocation 1: 1
94 For the sake of completeness, please describe the entire conventional treatment protocol
106 Describe the company and place of production .. among other things, describe in the introduction if it is the first time it is used in a clinical setting, if it is available on some app store, if it is open-source or open-access
Figure 2 is connected with line 94? Add timing, any repetition set.
138 shapiro is for normality, not for homogeneity between group..
143 Reliability?
Tables I don't understand what the p values refer to? when evaluating the TA of each group, when evaluating the intergroup differences at each timepoint? too confusing ....
Begin the discussion with the paraphrase of the goal. Then describe the major findings of the study. Avoid phrases like "The use of a telerehabilitation App is feasible." For three reasons, it is not a study result, also because the reliability has not been calculated, moreover it is too solid for a study with this sample, it lacks more than one bibliographic reference. What you can write is "appears to be, appears as ...", especially after fully describing the limitations of the manuscript.
Author Response
42 I suggest such a statement with reference to: "Telerehabiliation might provide specific immediate home rehabilitation services, guaranteeing continuous monitoring of patients and improving not only the state of health but above all the quality of life."
(ref: https://doi.org/10.1108/JET-11-2020-0047 )
RESPONSE: Similar idea can be found in the introduction (line 53).
53 I would suggest enriching the rationale with the lack of focus in the literature on home-based trunk therapy in people with stroke
RESPONSE: This idea was clarified in the introduction (line 60).
83 e Figure 1.. The fact that the enrollment is of 30 patients is a result, as well as the described flowchart is a result that led to the start of the study. To this continuation on line 79 add allocation 1: 1
RESPONSE: This information was entered into the manuscript (line 94)
94 For the sake of completeness, please describe the entire conventional treatment protocol
RESPONSE: The description of the usual care can be found in the methodology section (line 104)
106 Describe the company and place of production .. among other things, describe in the introduction if it is the first time it is used in a clinical setting, if it is available on some app store, if it is open-source or open-access
RESPONSE: Details of the App used in the study were included (line 119)
Figure 2 is connected with line 94? Add timing, any repetition set.
RESPONSE: The figure shows the exercises included in the App. The number of repetitions was introduced in the text (line 129)
138 shapiro is for normality, not for homogeneity between group.
RESPONSE: This bug has been rectified (line 158).
143 Reliability?
RESPONSE: This bug has been rectified
Tables I don't understand what the p values refer to? when evaluating the TA of each group, when evaluating the intergroup differences at each timepoint? too confusing .…
RESPONSE: The results referring to the intra-group and between-group comparison have been explained in all the tables.
Begin the discussion with the paraphrase of the goal. Then describe the major findings of the study. Avoid phrases like "The use of a telerehabilitation App is feasible." For three reasons, it is not a study result, also because the reliability has not been calculated, moreover it is too solid for a study with this sample, it lacks more than one bibliographic reference. What you can write is "appears to be, appears as ...", especially after fully describing the limitations of the manuscript.
RESPONSE: The paragraph has been rewritten according to the suggestions (line 262).
Round 2
Reviewer 1 Report
The authors successfully addressed major critical points. Considering that this is a preliminay trial, I suggest to specify this in the title.
Author Response
We greatly appreciate this opportunity and your suggestions. We have followed the last recommendation to outline the title of the manuscript.
Reviewer 2 Report
Given the opacity of the statistical methodology, the inappropriate description of the interventions and the unchanged presentation of the results, I am unfortunately inclined to recommend the unsuitability of the manuscript for publication.
96 15 and 15 is a result of the allocation, in methods section must describe the intervention design..
102 Still results of the subsequent enrollment.
108 As in the first revision it is not clear .. the intervention should be defined in detail in the interventions section. What duration? repetitions? how many exercises, did you have pauses?
109 BIAS.. If a randomized controlled trial is performed, these non-objectified intervention changes cannot be created .. at this point the results of the support of the experimental intervention are distorted
119 Farmalarm App (Inmovens Solution, Barcelona, Spain)
Figure 2: The list is 10 for each section exercises, the images are 8? Should the description be deepened? Was there a predefined order? How long for each exercise?
129 Was it possible for these patients to perform 10 repetitions for 30 exercises? 300 training sessions seem brutally infeasible to me
154 Among other things, compared to the first review, an intergroup evaluation is boasted, how was it conducted? An effect size calculation for each comparison at each timepoint?
Table 1 From what was said above at the baseline is there any significant difference between these groups? But above all, are the outcomes significantly different from the start?
Table 2 As for the first review, it is not clear what the intra-group p refers to, but above all what does a p mean for a comparison of intergroup averages T0-T1? did you make a mean difference comparison between the 2 groups? If so, why? but above all how?
Author Response
Thank you very much for considering this work for publication and the opportunity. Telerehabilitation is an important topic. In our opinion, this content will help to introduce telerehabilitation into the clinical practice of physiotherapy and rehabilitation, and will, on the other hand, facilitate future studies.
Next, we respond to their suggestions, which have greatly improved the writing of this paper.
96 15 and 15 is a result of the allocation, in methods section must describe the intervention design: Line 95 describes the randomization process
102 Still results of the subsequent enrollment: The changes have been made, considering this information as part of the results
108 As in the first revision it is not clear .. the intervention should be defined in detail in the interventions section. What duration? repetitions? how many exercises, did you have pauses? The details of the intervention have been entered (l.128)
109 BIAS. If a randomized controlled trial is performed, these non-objectified intervention changes cannot be created .. at this point, the results of the support of the experimental intervention are distorted In clinical practice, a rigid protocol with stroke patients is not feasible. Clinical recommendations should be followed respecting the abilities and needs of each individual. This idea is seen in several papers published in this journal (eg: Music Therapy Supports Children with Neurological Diseases during Physical Therapy Interventions, Gait Improvement in Chronic Stroke Survivors by Using an Innovative Gait Training Machine: A Randomized Controlled Trial, etc.). On the other hand, individuals with stroke should not be deprived of the best possible rehabilitation, which is why the usual care has been maintained in the two study groups.
119 Farmalarm App (Inmovens Solution, Barcelona, Spain): This information was entered on line 120
Figure 2: The list is 10 for each section exercise, the images are 8? Should the description be deepened? Was there a predefined order? How long for each exercise? The number of images does not coincide with the text because there were frames that were repeated, for example, the same exercise with the left and right upper limb. The figure has also been retouched to match the number of frames with the number of exercises in the App program.
129 Was it possible for these patients to perform 10 repetitions for 30 exercises? 300 training sessions seem brutally infeasible to me. The program is feasible to do in less than an hour. The exercise program was practiced in a face-to-face session as mentioned in line 137. Likewise, the patients should respect their level of fatigue and their abilities (eg, some participants could not perform some exercises). It is known that the exercise program must be fully customizable for each patient, but in this first phase of research and to reduce bias, it was decided to introduce the 32-exercise program in the App, available to all users.
154 Among other things, compared to the first review, and intergroup evaluation is boasted, how was it conducted? An effect size calculation for each comparison at each time point? For the intergroup analysis, the mean values and variances have been studied with the t-student test of independent samples (line 164).
Table 1 From what was said above at the baseline is there any significant difference between these groups? But above all, are the outcomes significantly different from the start? All the results are displayed in the tables. It can be seen that there are no significant differences between groups or in the variables at the beginning of the study, except for the gait study, in which the assessments are very scattered.
Table 2 As for the first review, it is not clear what the intra-group prefers to, but above all what does a p mean for a comparison of intergroup averages T0-T1? did you make a meaningful difference comparison between the 2 groups? If so, why? but above all how? The evolution in each group (intra-group) was studied, comparing the results between the different moments of data collection (paired samples). And the results between groups have been compared by comparing the differences obtained between one group and another (independent samples). The results tables were re-done for easier understanding.